# Evaluating the Within-Host Dynamics of *Ranavirus* Infection with Mechanistic Disease Models and Experimental Data

**DOI:** 10.3390/v11050396

**Published:** 2019-04-27

**Authors:** Joseph R. Mihaljevic, Amy L. Greer, Jesse L. Brunner

**Affiliations:** 1School of Informatics, Computing and Cyber Systems, Northern Arizona University, Flagstaff, AZ 86011, USA; 2Department of Population Medicine, University of Guelph, Guelph, ON N1G 2W1, Canada; agreer@uoguelph.ca; 3School of Biological Sciences, Washington State University, Pullman, WA 99163, USA; jesse.brunner@wsu.edu

**Keywords:** amphibian, *Ranavirus*, frog virus 3, mathematical models, Bayesian inference

## Abstract

Mechanistic models are critical for our understanding of both within-host dynamics (i.e., pathogen replication and immune system processes) and among-host dynamics (i.e., transmission). Within-host models, however, are not often fit to experimental data, which can serve as a robust method of hypothesis testing and hypothesis generation. In this study, we use mechanistic models and empirical, time-series data of viral titer to better understand the replication of ranaviruses within their amphibian hosts and the immune dynamics that limit viral replication. Specifically, we fit a suite of potential models to our data, where each model represents a hypothesis about the interactions between viral replication and immune defense. Through formal model comparison, we find a parsimonious model that captures key features of our time-series data: The viral titer rises and falls through time, likely due to an immune system response, and that the initial viral dosage affects both the peak viral titer and the timing of the peak. Importantly, our model makes several predictions, including the existence of long-term viral infections, which can be validated in future studies.

## 1. Introduction

Understanding the specific interactions between a host’s immune system and an infectious agent can be critical for understanding population-level patterns of infection prevalence and the dynamics of prevalence through time [1,2]. Mechanistic models of within-host dynamics use mathematical expressions to represent the processes of pathogen population growth and immune defenses that fight against pathogen growth. These types of models have revealed key insights, especially regarding the effects of various immune strategies on the evolution of pathogen virulence [3,4]. Within-host models also help us to understand the effects of chronic and acute infections on disease progression, pathogen evolution, and the links between within-host and among-host processes [5]. For example, Kennedy et al. [6] showed that models of within-host viral replication and immune processes accurately recapitulate the effects of viral exposure dose on the probability of death in an insect host. Here, we develop within-host models to understand the replication of ranaviruses in their amphibian hosts, and we compare our models’ expectations to laboratory data. 

Viruses in the genus *Ranavirus* can be impressively lethal, causing rapid host mortality and mass mortality events, especially in larval amphibians. However, the outcome of infection varies tremendously across host taxa [7,8] and life stage [9,10,11], as well as dose of exposure and numerous environmental factors (reviewed in Reference [12]). Much of this variation is presumably due to variation in the host-virus interaction, where viral replication is countered by immune responses. Indeed, some qualitatively distinct outcomes appear to have their basis in different immune responses. Adult *Xenopus laevis*, for instance, are refractory to ranavirus infections, while larvae often succumb, and this is due to their distinct immune responses [13,14]. The much more effective adult response to infection involves rapid, substantial CD8+ cellular responses, whereas tadpoles’ responses are dramatically less effective and slower. Larval *X. laevis* produce type I and type III interferon responses, which are effective at reducing viral replication, but are eventually overwhelmed by the virus [15,16]. However, while there has been great progress in elucidating the components of the immune system that interact with and help control ranavirus, many questions remain.

In particular, the dynamics of viral population growth are largely unknown outside of cell culture experiments. What are the dynamics of the virus population within the host? Does the host’s immune system effectively control the virus population, outside of the *Xenopus* model system? Does the immune response lead to viral clearance or can persistent infections remain? Does this outcome depend on the initial exposure dose? We propose that mathematical models of the within-host dynamics of ranaviruses, which provide a quantitative understanding of the host-virus interaction, will help answer these questions and will provide a means for integrating the numerous factors that appear to be important for the outcome of ranavirus infections. There are several additional advantages of using a mechanistic, model-based approach. First, parameterized disease models yield quantitative insights that can be validated by collecting new data. Second, we can simulate parameterized models under novel scenarios to predict how the system might behave. Third, we can use models to identify which novel sources of data would best inform our understanding under different scenarios. Thus, our approach offers a rigorous method to generate new hypotheses and guide future studies to understand the interactions between ranaviruses and host immune defenses.

To this end we set out to develop several versions of a mechanistic, within-host model that embodied hypotheses about the replication of frog virus 3 (FV3) and the host’s immune response to the infection. We then fit these models to a time series of FV3 titers within bullfrog (*Lithobates catesbeianus*) tadpoles that were experimentally infected with FV3. We used a model-comparison approach to determine the most parsimonious model(s) that could describe the observed viral titer through time and the effects of viral exposure dose. The salient features of these dynamics are: (1) a clear rise and then slower decline in viral titers over time, suggesting viral clearance by the host’s immune response and motivating our model-fitting exercise, (2) an earlier and higher peak in viral titers with higher exposure dose, and (3) low-level infections remaining detectable for seven weeks. We then use these models to generate testable hypotheses and guide future studies to understand the interactions between ranaviruses and host immune defenses.

## 2. Materials and Methods

Our goal is to test a set of hypotheses about how ranaviral populations replicate within their larval amphibian hosts and how the virus interacts with the larval host’s immune system. Our strategy is to develop mechanistic within-host growth models with various assumptions that correspond to our hypotheses. We then attempt to fit these models to serially-collected data on within-host titers, and we use a model-comparison approach to determine the most parsimonious model structure, and therefore the most parsimonious support for our hypotheses.

### 2.1. Experimental Assessment of Within-Host Ranaviral Dynamics

Details of the experiment can be found in Brunner et al. [17], but the essential details are that tadpoles (Gosner [18] stages 25–28) were exposed in 400 mL water baths for 24 h to one of three doses of the ranavirus FV3: a high dose (10^5^ plaque-forming units [pfu] mL^−1^; *n* = 150), medium-dose (10^3^ pfu mL^−1^; *n* = 150), or a low, but unknown dose (*n* = 90 in “mock” exposure to inadvertently contaminated cell culture media). The tadpoles were assigned to be euthanized and sampled on days 2, 4, 6, 8, 14, 21, 28, 35, 42, and 49 post exposure (*n* = 15 high-dose, *n* = 15 low-dose, and *n* = 9 “mock” tadpoles per time point), although some died and were sampled before their assigned date. Viral titers in liver and kidney samples were measured with a quantitative real-time PCR assay [19].

We do not include individuals that died before their pre-determined sampling date, because we focus on how the immune system is functioning against the virus, and we assume that individuals that died due to virus had immune systems that were overwhelmed. However, in the supplement (Appendix A), we demonstrate that our results are robust, and that including the dead individuals does not change the rank-order of our models or our general conclusions about the system.

### 2.2. Mechanistic Models of Within-Host Dynamics

We fit models to these time-series data that embodied two assumptions about viral replication and two about the host’s immune response. First, we fit models with exponential viral growth ($V^{\prime }=\phi V$) or with logistic growth [$V^{\prime }=\phi V\left(1-\frac{V}{K}\right)$]. In these differential equations, $V^{\prime }$ represents $\frac{dV\left(t\right)}{dt}$, where V is the density of virus within the host, ϕ is the per capita replication rate of the virus, and K is the viral carrying capacity. In some models, K can be viewed as the viral titer above which the virus kills the host [5]; however, because we are modeling mean-field dynamics, we do not explicitly account for virus-induced host mortality.

We then layer on possible immune system dynamics. From previous experimental work, we know that larval amphibians (at least *Xenopus*) show interferon type I and type III responses to FV3, although the latter appears more effective [15,16]. While we lack detailed knowledge of the immune response to FV3, especially in non-model amphibian species, we begin with the reasonable assumption that the production of the immune components, Z, responds to viral infection following the Michaelis-Menten form of enzymatic activity [20,21]:(1)$$Z^{\prime }=\psi Z\left(\frac{V}{V+\gamma }\right)$$

In this case, the engagement of the immune component ramps up to a maximum, ψZ, as the virus population increases. The rate of immune component production is mediated by γ, which is the half-saturation constant. In this formulation the density of immune components never returns to pre-infection levels. We therefore also considered a second formulation

(2)$$Z^{\prime }=\left(N_{Z}-\delta Z\right)+\psi Z\left(\frac{V}{V+\gamma }\right)$$

Here, $N_{Z}$ is the constant rate of production of the immune components, and δ is the per-capita background loss of the immune components. Moreover, we define $N_{Z}=\delta Z\left(0\right)$, such that, when no virus is present, the immune system stabilizes to a homeostatic level of immune component density equal to Z(0). In other words, we assume a balance of the immune components’ production and loss at equilibrium, when no virus is present. This means that there is a constant background level of immune components (Z(0)) in a host, and when a virus infects the host, the production of immune components ramps up until the virus is cleared.

Finally, we assume a mass-action attack rate of the immune system against the virus, which is a Type I Hollings’ functional response, such that our most complex model becomes:(3.1)$$V^{\prime }\;=\phi V\left(1-\frac{V}{K}\right)-\beta VZ$$

(3.2)$$Z^{\prime }\;=\left(N_{Z}-\delta Z\right)+\psi Z\left(\frac{V}{V+\gamma }\right).$$

We therefore model immune component production in response to virus infection as a fundamentally different process compared to the function of the immune component that limits viral population growth. We further note that all of our models are nested, and Table 1 shows all model formulations that we fit to our data set. We divide the models into two classes, A and B, which are differentiated by the assumptions about the immune system’s production dynamics. Class A models are mostly distinguished by the fact that, at equilibrium, if the immune system response is strong enough, the virus is driven extinct (i.e., full viral clearance). For Class B models, the virus is not driven extinct, such that the virus persists at low levels within the hosts (Appendix B).

On a less technical note, in some ways our models are not specific about how the immune system of the larval amphibian functions. For example, our model could equally represent a population of immune cells or a population of molecules generated by the immune system (e.g., interferon molecules) that act to control viral replication more or less indirectly. In other words, although we model a mass-action attack rate of the immune system against the virus, we do not necessarily mean that the virus and immune component must come into direct contact. This immune action could be indirect, for instance, by controlling apoptosis of virus-infected cells.

### 2.3. Fitting the Models to the Experimental Data

We employ a Bayesian approach to inference to fit the suite of dynamical models to the time-series of within-host viral titers. Then we use Bayesian information criteria to compare the within-sample predictive accuracy of our models, while penalizing model complexity. In this way, we seek to understand the most parsimonious model structure that captures the main features of the time series. All model-fitting code is available on our open-source Bitbucket repository (https://bitbucket.org/jrmihalj/ranavirus-within-host-dynamics/src/master/).

We use the open-source statistical programming language, Stan [22], to fit our differential equation models to the time-series data. Stan employs a Hamiltonian Monte Carlo (HMC) algorithm to sample from the model’s posterior. For each model, we sampled the posterior using three Markov chains, with a 2000 iteration warm-up, followed by 2000 iterations, for a total of 2000 samples recorded from each chain. We conducted various graphical and quantitative diagnoses of the chain behavior, including inspections for temporal auto-correlation, some of which are shown in the supplement (Appendix A). We assessed convergence with the Gelman-Rubin statistic, $\hat{R}$ [23]. All models converged after 4000 iterations.

Model dynamics are sensitive to parameter magnitudes. Therefore, in constructing our prior probability distributions for the parameters, we often restricted parameters to vague, but realistic ranges (see open-source model statements). All parameters were restricted to positive values, and for parameters that could take very large values (e.g., K) we set the parameter on a natural-logarithmic scale. To allow for potential correlations in our model parameters, we assumed that the main model parameters (i.e., ϕ,β,K,δ,ψ) had prior probabilities that were zero-truncated, multivariate normal, with estimated means and an estimated covariance matrix (based on underlying correlations). We used Cholesky Factorization to estimate the correlation structure, assuming vague priors for the correlations. Specifically, we used an LKJ prior on the correlations with a shape parameter equal to 2, which creates a correlation matrix closer to zero correlation (i.e., a conservative assumption) [22]. Additional details on prior structures and model specification can be seen in our open-source code repository. 

Besides estimating the model parameters, we also had to estimate the initial conditions of the system: The initial viral titer, V(0), and the initial immune component density, Z(0). As we have three viral dosage treatments, we estimated three independent V(0). We used a hierarchical prior structure, such that each treatment’s initial titer was drawn from a distribution with an estimated mean and variance (see model code). We further assumed that the initial immune component density, Z(0), did not vary among treatments. We therefore assumed that all larval amphibians start with the same average immune system component density that then responds to the invasion of the virus. As the magnitude of V(0) and Z(0) could be large, we again set these parameters on a natural-logarithmic scale.

The viral titer values spanned approximately six orders of magnitude. To improve numerical integration performance, by reducing the error in the integration algorithms due to such large viral titer values, we fit the differential equation models using natural log-transformed titer. Assuming that v=log(V) and z=log(Z), we can reformulate Equation (3) as follows:(4.1)$$v^{\prime }\;=\phi \left(1-\frac{e^{v}}{K}\right)-\beta e^{z}\;$$

(4.2)$$\;z^{\prime }\;=\left(\frac{N_{Z}}{e^{z}}-\delta \right)+\psi \left(\frac{e^{v}}{e^{v}+\gamma }\right).$$

Thus, the scaling of the parameters does not change, just the scaling of the state variables. This also allowed us to more reliably assume a Gaussian likelihood for log-transformed viral titer. We further allowed the residual variance to be different for each dosage treatment.

We also note that we only included non-zero titers in our data set for the model-fitting routine. Our reasoning is that a zero titer could represent an individual that never became infected, even upon exposure, or an individual that cleared infection later on. Because we could not distinguish between these two scenarios, we excluded these values. We did re-fit our models to the full data set, including zeros, to understand the effects on our inference. Importantly, this did not change the rank-order of our models using model comparison, but it generally made the models fit more poorly to the data, bringing down the average predicted titer and altering the marginal posterior estimates of the model parameters. We therefore do not consider these model fits further.

### 2.4. Model Comparisons

Our goal for model comparison was to determine the most parsimonious model(s) with the best goodness-of-fit. In practice, this means assessing the within-sample predictive performance of each model and correcting for model complexity. Given that we used a Bayesian approach to fitting our models to the data, we chose to use a Bayesian information theoretic approach to model comparison. Therefore, we calculated the leave-one-out information criterion (LOO-IC) for each model. Analogous to the Akaike information criterion (AIC), we can compare models based on the LOO-IC value, because they follow the deviance scale [24]. Thus, following the rule-of-thumb that ΔLOO-IC > 3 designates models with significantly different model performance, and where lower LOO-IC values indicate better-fitting, more parsimonious models.

## 3. Results

A full qualitative and statistical analysis of the experimental data can be found in [17]. For our purposes, it is important to note the clear rise and fall of viral titers over time in the medium and high dosage treatments, which suggested a possible immune response that was leading towards viral clearance, and which motivated our modeling-fitting and model comparison routine.

Model B2, our best-fitting and most parsimonious model, fits to the experimental data very well (Figure 1). This model includes exponential growth of the virus population, and an assumption of homeostatic levels of immune components within the host, and the model’s parameter estimates are shown in Table 2. Importantly, model B2 captures key features of the data that the other models failed to capture. First, the model correctly predicts that higher initial viral doses lead to higher peaks of within-host titer. Second, the model also shows that, with lower initial doses, the timing of the peak titer is delayed (Appendix A). In other words, it takes longer for the virus population to build up within the host. This is due, in part, to the exponential growth of the virus population and, in part, to the action of the immune system that inhibits viral replication.

Our model-fitting and model comparison approach suggests strong differences between the appropriateness of our different model structures. Models B2 and B3 had indistinguishable model fits (Table 1, Appendix A), but model B3 included an additional parameter, assuming logistic growth of the virus population (i.e., a carrying capacity within the host). This is a classic case of over-fitting, however, because the within-sample prediction for the model with more parameters was indistinguishable from the simpler model. Therefore, we conclude that model B2 is more parsimonious.

We had trouble fitting model A1 to the data, because it fits so poorly to the low-dose treatment. Therefore, the model ends up predicting a more or less linear decline in virus load across time, with high variance (Appendix A). This linear effect allowed the model to fit very well to the data from the low dose treatment, increasing its overall likelihood and improving its LOO-IC value. However, this parameterized version of the model fails to capture basic features of the data set. In contrast, model A2 fits to the data reasonably well (Appendix A). However, this model predicts the same peak titer across dosage treatments, leading to a very poor fit to the data from the low-dose treatment.

Model B1 also has a decent fit to the data (Appendix A). However, with this model’s estimated parameters, the dynamics show damped oscillations towards the endemic equilibrium. This pattern does not seem biologically realistic, as the data do not show any clear oscillations.

We do note that we had difficulty identifying the attack rate parameter β using model B2. This is likely because of the strong effects of viral replication rate ϕ and the growth rate of the immune components in response to viral density ψ on the overall model dynamics. Future experiments should be used to better estimate the attack rate parameter.

## 4. Discussion

Our study represents an initial step towards making model-based, quantitative predictions about ranavirus growth and immune system functioning within larval amphibian hosts, which complements the history of empirical work in these research areas. The experimental data with bullfrog (*Lithobates catesbeianus*) tadpoles suggests a non-monotonic relationship between within-host viral titer and time post-exposure. That is, an immune response appears to be reducing the growth of the virus over time. This is broadly consistent with the immunological studies with *Xenopus laevis* tadpoles, showing a robust interferon response [15,16]. Through model comparisons, we identified a model of exponential viral growth and a mass-action immune system response that captures key features of our data and generally fits our data quite well. Importantly, this model-based approach allows us to make predictions and to develop testable hypotheses for follow-up experiments. However, we did not measure immune responses directly in our experiment, and therefore cannot conclude with certainty that the patterns we observe in viral titer are driven by immune system processes. To clarify these mechanisms, below we suggest specific ways to collect and analyze data in future studies.

The best-fitting model predicts that individuals that survive infection should exhibit a viral titer pattern that rises and falls over time. This prediction contrasts with the empirical work that uses *Xenopus*, because larval *Xenopus* almost always die from frog virus 3 (FV3) exposure. Even a pre-treatment of type III interferon, which should boost the immune response, only delays mortality in the *Xenopus* system [16]. Unfortunately, we do not know of any studies that have tested whether similar host immune responses exist in our study organism, the American bullfrog. However, it is well known that the American bullfrog is uncommon in its high resistance and tolerance to multiple diseases, including FV3 [7,25] and the amphibian chytrid fungus (*Batrachochytrium dendrobatidis*) [26,27]. Therefore, it is not surprising that we could infer a more robust immune response from this species compared to *X. laevis*. Still, our current data cannot validate that, in surviving individuals, virus loads rise and then fall, because we do not have individual host-level time series of viral load. Future studies could use various sources of eDNA (e.g., swabbing, skin scrapes, or water filtration), which are correlated with internal titers [17], to validate whether there is a rise and fall of viral load in individuals. Alternatively, it may be fruitful to experiment with larger-bodied animals that would permit sampling blood repeatedly through time. 

Our model also predicts that, if a host does not die from the initial exponential growth of the virus, the immune system will decrease the virus to an endemic equilibrium (Appendix B). In other words, we should see persistent infections with very low levels of virus in some infected bullfrog tadpoles. It is worth noting that this model prediction is not relevant to quiescent virus persisting in particular tissues, such as that found in peritoneal leukocytes in adult *Xenopus* [28]. Rather, the persistence in our model is due to a dynamic balance between viral replication and host immune responses. In any case, our model predicts low-level infections that would be detectable beyond 60 days from initial exposure time and should be actively replicating. That said, there are also several reasons that an endemic equilibrium may not hold. First, demographic stochasticity in the virus population could cause low-level infections to fade out [6,29]. Second, variability in the immune response over time (e.g., waning immunity) could cause fluctuations in viral titer. Third, the costs of continued immune responses could lead to host death even with low levels of infection [30].

Several of the tested model structures predict full viral clearance at equilibrium, but none of these models fit as well to the observed dynamics in bullfrog tadpoles. Future studies that collect time-series data and that have some long-term exposures (e.g., greater than 50 days) will improve our ability to distinguish between viral clearance and viral persistence. Furthermore, because we did not have simultaneous empirical data from the larval immune system, our model fitting algorithm was agnostic to the model’s prediction of immune system dynamics. If future studies simultaneously capture time-series data on viral titer and measures of larval immune system, we would have much more power to distinguish between different models of within-host dynamics.

There are key features of the data set that our deterministic modeling scheme does not fully capture, and these are areas for future model development. First, the main process that our model ignores is host mortality due to viral growth. For instance, our current model predicts that immunocompromised individuals would show exponential viral growth, and in reality, this likely lead to death. Future versions of our model could include stochasticity, following Kennedy and Dwyer [6], such that demographic stochasticity in the viral population could lead to the virus exceeding a threshold growth rate or titer that causes host death. Then, we could analyze the probability of this occurring over many realizations of the stochastic model. This would allow us to explicitly evaluate how initial exposure dosage affects the probability of host mortality, from a within-host, mechanistic framework. 

Viral titers within a host can be important determinants of the infection dynamics among hosts [3], affecting both the duration of infections and their propensity to be transmitted. Many dose-response experiments, for instance, show that hosts exposed to higher doses of ranaviruses, and so presumably harbor larger viral populations, are more likely to die and die more rapidly [31,32,33]. More intensely infected individuals may also shed more virus (e.g., Reference [17]) and thus be more infectious. Similar effects of the intensity of infection are common in other host-virus systems [34,35], as well as for other amphibian pathogens (e.g., *B. dendrobatidis*; [36]). Pathogen titers or intensity of infection can be a natural way in which to link within and between-host dynamics [3,37,38]. Within-host models of host-virus interactions may therefore be a fruitful way of integrating the myriad effects that are known to influence the outcome of ranavirus infections. 

## Figures and Tables

**Figure 1 viruses-11-00396-f001:**
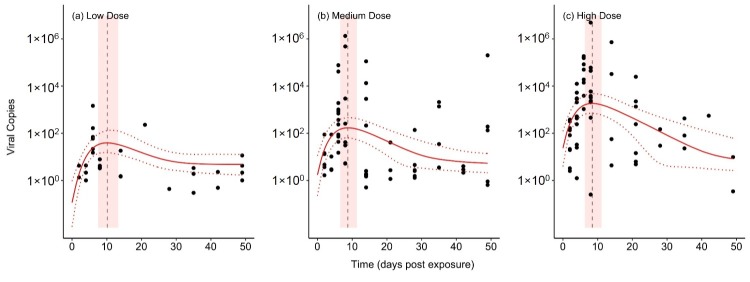
The fit of model B2 to the experimental data. Circles are data points representing the viral DNA copies from individual bullfrog tadpoles that were sampled on a given day. The median model fit (solid red line) and 95% Bayesian credible interval (CI) of the fit (dashed red lines) are shown. Additionally, the median (dashed vertical line) and 95% CI (light red polygon) are shown for the time of the maximum viral titer predicted by the model.

**Table 1 viruses-11-00396-t001:** Model structures and model comparisons. The bolded model (B2) is the most parsimonious based on LOO-IC selection and the lower number of parameters compared to B3.

Class	ID	Structure	Notes	Penalty (*p*LOO)	LOO-IC	ΔLOO-IC
A	A1	$V^{\prime }=\phi V-\;\beta VZ$ $Z^{\prime }=\psi Z\frac{V}{\left(V+\gamma \right)}$	Drives virus extinct. Z goes to equilibrium $Z_{\left(\infty \right)}$, which is above $Z_{\left(0\right)}$.	8.5	782.9	15.5
	A2	$V^{\prime }=\phi V\left(1-\frac{V}{K}\right)-\;\beta VZ$ $Z^{\prime }=\psi Z\frac{V}{\left(V+\gamma \right)}$	Conditions under which virus goes to carrying capacity. Or virus goes extinct.	7.1	811.7	44.3
B	B1	$V^{\prime }=\phi V-\;\beta VZ$ $Z^{\prime }=(N_{Z}-\delta Z)+\;\psi ZV$	Damped oscillations to a stable point equilibrium, where virus is persistent in host. The model fit shows several oscillations before equilibrium.	12.7	826.6	59.2
	**B2**	$V^{\prime }=\phi V-\;\beta VZ$ $Z^{\prime }=(N_{Z}-\delta Z)+\;\psi Z\frac{V}{\left(V+\gamma \right)}$	Spike in viral load, then decline to stable point equilibrium, where virus is persistent in host.	**10.1**	**768.6**	**1.2**
	B3	$V^{\prime }=\phi V\left(1-\frac{V}{K}\right)-\;\beta VZ$ $Z^{\prime }=(N_{Z}-\delta Z)+\;\psi Z\frac{V}{\left(V+\gamma \right)}$	Over-fitting. Extra parameter (carrying capacity, *K*) unnecessary.	9.1	767.4	0

**Table 2 viruses-11-00396-t002:** Parameter estimates (median and 95% credible intervals) from the most parsimonious model, B2.

Parameter	Description	Units	Estimate
V(0) low	Initial viral densities (per dosage)	Viral DNA copy (VC)	0.12 (0.01–0.89)
V(0) medium			1.47 (0.24–11.55)
V(0) high			24.10 (5.31–146.85)
Z(0)	Initial immune component denisty	Immune component (IC)	0.35 (0.04–4.43)
ϕ	Viral replication rate	day^−1^	2.39 (1.07–4.63)
β	Mass-action attack rate	(IC)^−1^ day^−1^	1.75 (0.15–6.28)
$N_{Z}$	Rate of production that ensures return of immune system to homeostasis	(IC) day^−1^	$N_{Z}=\delta Z\left(0\right)$
δ	Rate of decline that ensures return of immune system to homeostasis	day^−1^	1.29 (0.41–3.92)
ψ	Immune component growth rate in response to virus	day^−1^	0.99 (0.19–3.56)
γ	Half saturation constant	VC	0.13 (0.02–1.02)

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
