# Peer review of "Evaluating the Within-Host Dynamics of *Ranavirus* Infection with Mechanistic Disease Models and Experimental Data"

_viruses, 2019, doi:10.3390/v11050396_

Round 1

Reviewer 1 Report

See attached file.

Reviewer 2 Report

This manuscript describes the comparison of mechanistic models that predict viral infections against derived viral loads from bullfrog tadpoles infected with FV3. Out of several applied models, the authors choose one that best describes their data across low, medium and high FV3 infection doses and over the course of 50 days of infection. 

This is a well-written manuscript, with interesting results and a promising model that fits the empirical data rather nicely. 

Please find my suggestions on how to improve this manuscript below: 

Throughout the manuscript, the authors describe their models as a measure of viral growth relative to the host immune response. While the authors present very compelling FV3 load data that supports the predictions of their B2 model, they do not present experimental data confirming that the tadpole immune response is the cause of the changes in the FV3 loads. I suggest that that authors either present experimental data linking the tadpole immune response (maybe interferon) to the changes in the viral loads or make changes to their text that acknowledges the fact that there is no certainty that the observed changes in the FV3 loads are due to the tadpole immune response as opposed to other physiological changes or the nature of the viral replication strategy within this host. 

Does B2 (or the other models) take into differences in tadpole versus adult frog immune responses and/or a susceptible versus resistant amphibian host?

FV3 (and presumably other ranaviruses) may persist past acute infections as latent reservoirs. At present it is not know whether this is a function of the frog immune system since not all infected animals establish these latent reservoirs. The B2 model predicts that there is a residual FV3 reservoir that persists as the result of a balance between the host immune system and the virus replication. However, not all viruses form such reservoirs and it is presently not clear what the determinants of FV3 persistence are.  Is this aspect of the model reliable as a predictor of virus infection outcomes, or is this a coincidence that the model reflects current hypotheses.

Please change the description of the immune response from “growth of the immune components, etc” to “mounting an immune response” or “engaging an immune response”. 

Similarly, please change “viral growth” to “viral replication”.

Author Response

This manuscript describes the comparison of mechanistic models that predict viral infections against derived viral loads from bullfrog tadpoles infected with FV3. Out of several applied models, the authors choose one that best describes their data across low, medium and high FV3 infection doses and over the course of 50 days of infection. 

This is a well-written manuscript, with interesting results and a promising model that fits the empirical data rather nicely. 

Please find my suggestions on how to improve this manuscript below: 

Throughout the manuscript, the authors describe their models as a measure of viral growth relative to the host immune response. While the authors present very compelling FV3 load data that supports the predictions of their B2 model, they do not present experimental data confirming that the tadpole immune response is the cause of the changes in the FV3 loads. I suggest that that authors either present experimental data linking the tadpole immune response (maybe interferon) to the changes in the viral loads or make changes to their text that acknowledges the fact that there is no certainty that the observed changes in the FV3 loads are due to the tadpole immune response as opposed to other physiological changes or the nature of the viral replication strategy within this host. 

We appreciate the reviewer’s criticism. Unfortunately, no concurrent data on tadpole immune function was recorded in this experiment. Moreover, we don’t know of any data that concurrently measures viral titer and immune function over time, and therefore we cannot make such a comparison in this study. We do however advocate that these data be collected in the future.

We are now more explicit about the fact that we did not collect concurrent immune function data, and we can therefore not rule out alternative hypotheses about why virus declines over time.

Related changes to the text:

Lines (352): “However, we did not measure immune responses directly in our experiment, and therefore cannot conclude with certainty that the patterns we observe in viral titer are driven by immune system processes. To clarify these mechanisms, below we suggest specific ways to collect and analyze data in future studies.”

Does B2 (or the other models) take into differences in tadpole versus adult frog immune responses and/or a susceptible versus resistant amphibian host?

Our models are built to include hypothetical mechanisms of immune function that are informed by experiments with Xenopus larvae. However, we do not include specific immune mechanisms that represent the specific pathways of immune function in either adult or larval frogs. In other words, our representation of the larval immune system is agnostic to the mechanisms that occur (e.g., cell-killing, direct attack of the virus, etc.)

FV3 (and presumably other ranaviruses) may persist past acute infections as latent reservoirs. At present it is not know whether this is a function of the frog immune system since not all infected animals establish these latent reservoirs. The B2 model predicts that there is a residual FV3 reservoir that persists as the result of a balance between the host immune system and the virus replication. However, not all viruses form such reservoirs and it is presently not clear what the determinants of FV3 persistence are.  Is this aspect of the model reliable as a predictor of virus infection outcomes, or is this a coincidence that the model reflects current hypotheses.

We agree that hosts that are viral reservoirs and the mechanisms that lead to reservoir status are complex. We do not know whether our model reliably predicts these dynamics in the subset of American bullfrog larvae that we study here, but we believe that this deserves further research. Indeed, we comment upon this in the discussion (lines 374), about quiescent virus.

Please change the description of the immune response from “growth of the immune components, etc” to “mounting an immune response” or “engaging an immune response”. 

We have made the requested change throughout.

Similarly, please change “viral growth” to “viral replication”.

We have made the requested change throughout.

Round 2

Reviewer 1 Report

In their revised manuscript ``Evaluating the within-host dynamics of Ranavirus infection with mechanistic disease models and experimental data'', Mihaljevic et al. did a really nice job of responding to my questions from the previous review. I appreciate their time and effort.  At this point, I only have a couple of minor suggestion. 

1. In a number of places in the manuscript the authors refer to their open-source code repository for details on subjects like prior structure and model specification.  I feel that having the information in a supplemental or a data repository rather than on a coding repository such as bitbucket or github is a better option.  Given the quick switch from github to bitbucket, I wonder how realistic it is to expect these repositories to maintain their current status among academia and be a long-term storage site for this information.  Dryad or other repositories seem more fit for this.  

2. The other comment is with regards to figure 1.  In the caption, the authors write, ''Also, the median (dashed vertical line) and 95\% CI (light red polygon) are shown for the time of the maximum viral titer predicted by the model.'' In my version of the manuscript, I couldn't find the polygons or the dashed vertical line.

That said, overall, the authors have written a very nice manuscript.

Author Response

In their revised manuscript ``Evaluating the within-host dynamics of Ranavirus infection with mechanistic disease models and experimental data'', Mihaljevic et al. did a really nice job of responding to my questions from the previous review. I appreciate their time and effort.  At this point, I only have a couple of minor suggestion.

1. In a number of places in the manuscript the authors refer to their open-source code repository for details on subjects like prior structure and model specification.  I feel that having the information in a supplemental or a data repository rather than on a coding repository such as bitbucket or github is a better option.  Given the quick switch from github to bitbucket, I wonder how realistic it is to expect these repositories to maintain their current status among academia and be a long-term storage site for this information.  Dryad or other repositories seem more fit for this. 

We are open to depositing our data and code in Dryad, as well as our open-source repository on Bitbucket. However, for Dryad, our manuscript must be formally accepted before they allow data upload. Therefore, we pledge to upload our data and code on Dryad after acceptance.

2. The other comment is with regards to figure 1.  In the caption, the authors write, ''Also, the median (dashed vertical line) and 95\% CI (light red polygon) are shown for the time of the maximum viral titer predicted by the model.'' In my version of the manuscript, I couldn't find the polygons or the dashed vertical line.

We apologize, and we’re not sure why it is not showing up on the reviewer’s version of the manuscript. It seems to show up on the .docx file but not the .pdf file. Therefore, we uploaded the figure separately just in case.

That said, overall, the authors have written a very nice manuscript.